# Admixture of Holothurian Species in the Hellenic Seas (Eastern Mediterranean) as Revealed by RADseq

**Georgios A. Gkafas** [1,*] , **Joanne Sarantopoulou** [1] , **Chrysoula Apostologamvrou** [1] , **Chryssanthi Antoniadou** [2] , **Athanasios Exadactylos** [1] , **Georgios Fleris** [3] **and Dimitris Vafidis** [1]

[1] Department of Ichthyology and Aquatic Environment, University of Thessaly, Fytokou Str., 38446 Volos, Greece; saradopo@uth.gr (J.S.); chapostol@uth.gr (C.A.); exadact@uth.gr (A.E.); dvafidis@uth.gr (D.V.)

[2] Department of Zoology, School of Biology, Aristotle University, 54124 Thessaloniki, Greece; antonch@bio.auth.gr

[3] Directorate of Fishing Activity and Product Control, Directorate General of Fisheries, Hellenic Ministry of Rural Development and Food, Andrea Syngrou Avenue 150, Kallithea, 17671 Athens, Greece; fleris@hotmail.com

[*] Correspondence: gkafas@uth.gr; Tel.: +30-2421-093145

**Abstract:** Admixture and hybridization may play a key role in population dynamics and speciation with respect to habitat, demographic history, and adaptive selection. The present study examines the genetic diversity of two congeneric—but in different subgenera—holothurians that live in sympatry in mixed populations. Strong evidence of admixture was provided by analyzing RAD sequencing data from 90 and 67 individuals of *Holothuria* (*Holothuria*) *tubulosa* and *Holothuria* (*Roweothuria*) *poli*, respectively, from various areas of the Hellenic Seas (eastern Mediterranean). Coalescent demographic analysis revealed a recent directional gene flow from *H. poli* to *H. tubulosa*. The two species populations diverged around 13.5 thousand years ago, just after the Last Glacial Maximum. According to the pairwise sequential Markovian coalescent approach, the historical population effective sizes for both species declined during the last Pleistocene glaciations, probably due to population decline, followed by a relative rapid recovery as it is calculated using LD methods. The presented results imply a role for admixture upon secondary contact and are consistent with the recent suggestion that the genomic underpinning of ecological speciation often has an older, allopatric origin.

**Keywords:** holothurians; RADseq; admixture; hybridization; pairwise sequential Markovian coalescent; gene flow; species coexistence

## 1. Introduction

Population dynamics of marine species are subject to evolutionary processes driven by geological climate changes. Such changes occurred in the Quaternary, where significant alterations in species distribution and structure were evident (e.g., [1–3]). One of the most important geological periods was the Pleistocene glaciations, where, due to changes in oceanographic features, barriers to gene flow [4,5] and isolated refugia [6] were created. These changes in habitat availability affected historical population effective sizes and possibly drove profound allopatric speciation [7].

Hybridization between different taxa is widespread across distinct phylogenetic lineages [8–10]. Such evolutionary processes occur between parapatric species distributed in common site fidelity areas [11]. Interbreeding between distinct lineages as a result of the admixture of potential isolated gene pools may cause a loss of the genetic signature due to introgression [12], which, depending on the level, may result in speciation [13]. As a potential result, the loss of ancestral alleles due to changes in the population effective size along with admixture may be the result of contemporary ecological divergences and possible reproductive barriers [14]. To this extent, Bernatchez and Dodson [15] argue that sympatric

species have been found to share the same ecological niches favoring allopatry (e.g., striped and common dolphins in enclosed gulfs—[16]). Moreover, genetic admixture events are capable of ecological speciation, resulting in complex population dynamics [17,18]. However, due to this complex coexistence of sympatric and prezygotic isolated species, it is rather challenging to unravel a given divergence mechanism. Nevertheless, such derived speciation of sympatric species may be influenced by selection forces on the adaptive mechanisms [19], resulting in a partial divergence with limited ongoing genome-wide gene flow [20].

Holothurians constitute a rather unique group of marine animals from an evolutionary perspective and are highly divergent compared to other extant echinoderm classes [21]. Despite the intensification of relevant research during the last decade, the understanding of their systematics and phylogenetic evolution remains fuzzy [22]. The type family of the class, Holothuriidae, includes the type genus *Holothuria*, which is by far the most diverse, with over 80% of the described species. This genus includes 18 subgenera, among which *Holothuria* and *Roweothuria* are distributed in the Mediterranean Sea, often as prominent sympatric species in sedimentary marine benthic habitats [23,24]. In the Hellenic Seas, the most widely distributed *Holothuria* species are *Holothuria* (*Holothuria*) *tubulosa* Gmelin, 1793, and *Holothuria* (*Roweothuria*) *poli* Delle Chiaje, 1824, which are intensively harvested as gastronomic delicacies for human consumption [25,26]. Both species follow a patchy pattern of distribution in the Hellenic Seas, with locally dense coexisting populations in proximity with extremely sparse or null populations [25,27,28]. Habitat heterogeneity and trophic niche segregation have been proposed as the most probable underlying mechanisms to explain their coexistence in other Mediterranean areas [24]. As closely related congeneric species with broadcast spawning behavior [29,30], they may be susceptible to hybridization, despite being classified to distinct paraphyletic subgenera [22], especially in areas with dense sympatric populations, leading to sea cucumbers with mixed morphological features [31].

Previous molecular studies of *H. tubulosa* show a rather strong genetic differentiation between semi-enclosed areas in the eastern Mediterranean [32]. Signatures of historical population demography show a bottleneck event for the species approximately after the Last Glacial Maximum (LGM), revealing a massive loss of genetic diversity, followed by a more recent recovery of the population in the area. For *H. poli*, unfortunately, there is no such data available in the bibliography, despite the recent analysis of its complete mtDNA genome sequence [31]; however, a similar ecological behavior may be inferred. *H. poli* is considered among the most divergent species of the family [22].

Considering all the above, the present work aims to assess the genetic diversity of *H. tubulosa* and *H. poli* in the Hellenic Seas and provide evidence of possible admixture of potential isolated gene pools of the two species. Various demographic models were examined under different gene flow scenarios between the two lineages to better explain any potential admixture of the coexisting populations in the study area.

## 2. Materials and Method
### 2.1. Samplings

Specimens of *H. tubulosa* and *H. poli* were collected by diving up to 25 m from nine different geographical areas of the Hellenic Seas (Figure 1) during summer–autumn from 2019 to 2021. Overall, 180 sea cucumbers (110 specimens of *H. tubulosa* and 70 of *H. poli*) were obtained and processed. Right after collection, each individual was dissected with high-precision anatomic tools, tissues were immediately placed in DNase- and RNase-free Eppendorf tubes and thereafter frozen in liquid nitrogen. The frozen samples were transferred to the laboratory, where they were stored at −80 °C for genetic analysis.

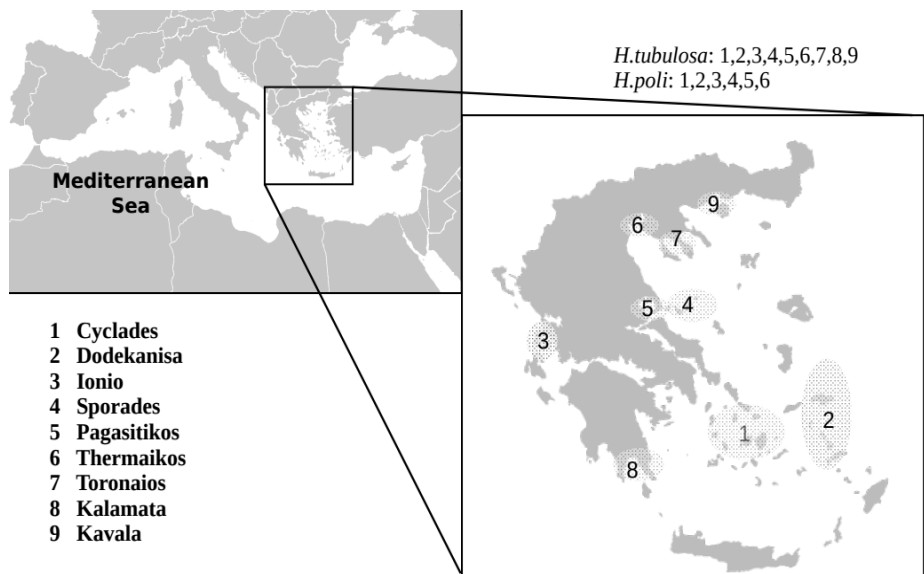

**Figure 1.** Map of the Hellenic Seas indicating the sampling areas of the two holothurian lineages. The number of *H. tubulosa* specimens per sampling area is 14 from Cyclades, 13 from Dodecanese, 4 from Ionian, 17 from Sporades, 8 from Pagasitikos (8), Thermaikos (7), Toronaios (7), Kalamata (9), and Kavala (11). *H. poli*: Cyclades (13), Dodekanisa (17), Ionio (4), Sporades (17), Thermaikos (6), and Toronaios (10).

### 2.2. Molecular Procedures

2.2.1. DNA Extraction and Sequencing

Muscle tissue (longitudinal muscle bands) was used for DNA extraction using the PureLinkTM Genomic DNA Kit (Invitrogen, Carlsbad, CA, USA). A DNA library was constructed following the ddRADseq protocol described in Peterson et al. [33]. We chose a 6 bp cutter (HindIII) and a 4 bp cutter (MspI) based on in silico simulations with the R package SimRAD [34]. The fragment size selection window was 300–500 bp. Sequencing was paired-end (2 × 151 bp) in one lane on an Illumina NextSeq_500 (Illumina, San Diego, CA, USA). Reads were trimmed to 110 bp and demultiplexed using the process_radtags command of the software STACKS v.2.64 [35].

2.2.2. Mapping, Variant Calling, and Filtering

Paired reads were mapped against the *Holothuria glaberina* genome [36] for both species using BWA v. 0.7.12 [37]. Each resulting sam file was converted to bam format using SAMtools v. 1.3. [38]. Using the command SelectVariants, indels and non-biallelic single nucleotide polymorphisms (SNPs) were filtered out. Then, using the command VariantFiltration, SNPs were filtered based on mapping quality using the following settings: –filterExpression 'QD < 2.0||FS > 60.0||MQ < 30.0||MQRankSum ← 12.5||ReadPos-RankSum ← 8.0'. The QUAL score (QD) was normalized by allele depth for a variant, and the Phred-scaled $p$-value (FS) used Fisher's exact tests to detect strand bias. The MQRankSum command set the $z$-score from a Wilcoxon rank-sum test of Alt versus Ref read mapping qualities, and ReadPosRankSum performed this for reading position bias. Loci were assembled using the GATK HaplotypeCaller [39].

VCF files were further filtered based on both genotype quality and depth of coverage greater than 5 using VCFTools v.4.2 [40], as well as to retain only SNPs genotyped in at least 80% of the individuals. Following that, SNPs with a depth of coverage twice the mean depth of coverage were discarded to remove potentially paralogous loci. All individuals with more than 20% missing data were removed, and only variants with MAF greater than 0.05 were retained. Finally, the software PLINK v. 1.9 [41] was used to filter out SNPs showing significant departures from Hardy–Weinberg equilibrium with an alpha level of 0.05 as well as to prune out putatively linked loci using an r2 threshold of 0.5.

2.2.3. Genomic Analysis

Genetic diversity indices (number of alleles and number of polymorphic sites) and estimates of heterozygosity ($H_{OBS}$, $H_{EXP}$, and $F_{IS}$) were performed using the program SAMBAR [42] through the R platform [43], following authors recommendations. We used the program Stampp to generate Weir and Cockerham [44] $F_{ST}$ estimates, along with associated significance values.

Population structure was assessed using the faststructure software for optimal assignment of the analyzed individuals, aiming to estimate the K value, reflect the theoretically most likely number of populations [45], and assume correlated allele frequencies and admixture. Three independent replicates were run for each of K (1 < K < 6). Following test runs, the burn-in length and length of simulation were set at 1,000,000 and 3,000,000 repetitions, respectively. Stucture Harvester [46] was used to assess the likelihood value of the different K values and to implement ΔK method [47] reflecting the highest hierarchical level of structuring. PCA analysis through PLINK v.1.9 [41] was further assessed for individual genotypes clustering through eigen values.

For the detection of admixture, the software ADMIXTURE v.1.3.0 [48] was implemented to detect clusters and assign individual-level ancestry proportions from each cluster. Also, the model-based approach, as implemented in NEWHyBRIDS [49], was used to estimate, for each sample, the posterior probability of it belonging to either *H. tubulosa* or *H. poli* category. The number of iterations was set to 1,000,000, with 25% as a burn-in.

The software GONE v.1.0 [50] was used to obtain contemporary estimates of the effective population size using Linkage Disequilibrium (LD). The vcf file was converted to ped and map format using the PLINK v.1.9 software [41]. The parameters for the analysis were set as follows: number of generations was 2000, number of bins was 400, no MAF pruning was applied, and the maximum value of recombination rate (c) was set to 0.05. The number of internal replicates was set to 40 in order for the program to provide the geometric mean of the consensus estimate of the historical Ne out of these replicates. On the other hand, for the estimation of the historical population effective size, the software PSMC [51] was used using the pairwise sequential Markovian coalescent approach. Aligned mapped reads (bam files) were converted to consensus sequences in FASTQ format using samtools/bcftools pipeline through the vcfutils.pl script. Mapped reads were filtered for a minimum mapping quality (q) of 10 and a minimum base quality (Q) of 10. The minimum (d) and maximum (D) coverage was calculated to allow for vcf2fq, and it was set to 1 and 15, respectively ($-$d value to a third of the average depth and $-$D value to twice). For the final PSMC command, we used 64 atomic time intervals and 28 (=1 + 25 + 1 + 1) free interval parameters. Finally, the PSMC plot was drawn using the command 'psmc_plot.pl' with the per-generation mutation rate '$-$u' and the generation time in years '$-$g' set to $1 \times 10^{-8}$ and 3, respectively, according to Zhang et al. [21].

To estimate divergence times of *H. tubulosa* and *H. poli* lineages, a demographic analysis was implemented using the coalescent simulator fastsimcoal2 [52]. VCF files were converted to site frequency spectra (SFS) using the easySFS program [53] to fit model parameters to the observed data by performing coalescent simulations. Generation interval for both species was set to 3 [21]. Five isolation and migration models were examined under different scenarios of gene flow (Figure 2). In the first model, early gene flow after divergence was assumed, without recent gene flow as reproductive isolation becomes stronger (Scenario 1). In the second model, an early gene flow after divergence was assumed following a lower gene flow as reproductive isolation accumulates (Scenario 2). In the third model, a constant gene flow from divergent time until now was set (Scenario 3). The fourth model dedicated a gene flow to recent times without any early gene flow after divergence (Scenario 4), and the fifth model assumed no gene flow at all between the two lineages (Scenario 5).

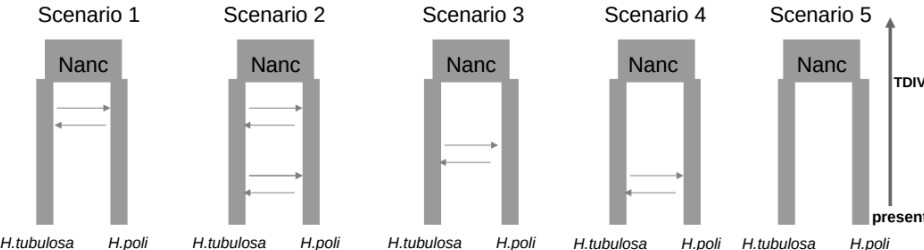

**Figure 2.** Demographic models of divergence times ($T_{DIV}$) and different gene flow scenarios (gray arrows) between the *H. tubulosa* and *H. poli* lineages. Scenario 1: early gene flow; Scenario 2: early and lower recent gene flow; Scenario 3: ongoing gene flow; Scenario 4: recent gene flow; Scenario 5: no gene flow. Nanc: ancestral population.

## 3. Results

### 3.1. SNPs Calling

After quality control (samples with less than 1 million reads were rejected), 90 samples for *H. tubulosa* and 67 samples for *H. poli* were retained. An average of 107,449 quality trimmed sequence reads were generated, from which we called 14,221,157 raw SNPs for *H. tubulosa*. The application of the filtering criteria described in the Materials and Methods section resulted in a final dataset of 4253 high-quality filtered SNPs. As for *H. poli*, an average of 100,736 quality trimmed sequenced reads were retained. A total of 10,744,004 raw SNPs were generated, and after filtering, a final panel of 4758 was used. (Supplementary Table S1 for a downstream of applied filtering and remaining SNPs).

### 3.2. Genetic Differentiation and Diversity

Although both species show relatively low values of expected heterozygosity, no significant levels of inbreeding were estimated (Table S2). For *H. poli*, the geographical area of Thermaikos shows a relatively low number of polymorphic sites, whereas for *H. tubulosa*, Ionio and Kalamata show a lower number of polymorphic sites. To test for genetic differences, pairwise $F_{ST}$ values were evaluated (Table 1). These were highly significant for comparisons involving Pagasitikos and Thermaikos gulfs ($F_{ST}$ = 0.110, $p < 0.001$) for *H. tubulosa*. In general, the population from Thermaikos showed significant differentiation in most pairwise comparisons, except for Ionio, Kavala, and Sporades. On the contrary, the Pagasitikos Gulf, the other semi-enclosed gulf, did not show any significant differentiation with the other geographical areas. No significant differentiation was detected among the offshore populations of the species as well. In the case of *H. poli*, no significant pairwise differentiations were detected, with the Sporades and Cyclades island complex being the only exception.

**Table 1.** $F_{ST}$ pairwise differences for *H. tubulosa* and *H. poli* geographical regions. CYCL: Cyclades, DOD: Dodekanisa, ION: Ionio, KAL: Kalamata, KAV: Kavala, PAG: Pagasitikos, SPO: Sporades. Bold indicates significance.

| Species | | CYCL | DOD | ION | KAL | KAV | PAG | SPO |
|---|---|---|---|---|---|---|---|---|
| | DOD | 0.002 | | | | | | |
| | ION | 0.0005 | 0.008 | | | | | |
| | KAL | 0.004 | 0.005 | 0.006 | | | | |
| *H. tubulosa* | KAV | 0.002 | 0.003 | −0.002 | 0.005 | | | |
| | PAG | 0.004 | 0.008 | −0.001 | **0.014** | 0.008 | | |
| | SPO | 0.003 | 0.003 | −0.003 | 0.003 | −0.001 | 0.001 | |
| | THE | **0.011** | **0.011** | 0.0006 | **0.025** | 0.006 | **0.015** | 0.007 |
| | | CYCL | SPO | DOD | ION | | | |
| | SPO | 0.011 | | | | | | |
| *H. poli* | DOD | 0.004 | 0.004 | | | | | |
| | ION | 0.006 | −0.003 | 0.003 | | | | |
| | THE | 0.005T | 0.006 | 0.002 | 0.003 | | | |

PCA and structure analysis show an apparent panmixia for both species in the study area (Figure 3). LnPD values leveled off after K = 3 (Supplementary Information, Table S3). Accordingly, they do not reveal any clear geographical pattern of differentiation for *H. tubulosa*. In the case of *H. poli*, LnPD and ΔK values revealed two prominent clusters, although without a clear-cut pattern. PCA analysis conducted for both species showed no differentiation between the two lineages (Supplementary Information, Figure S1).

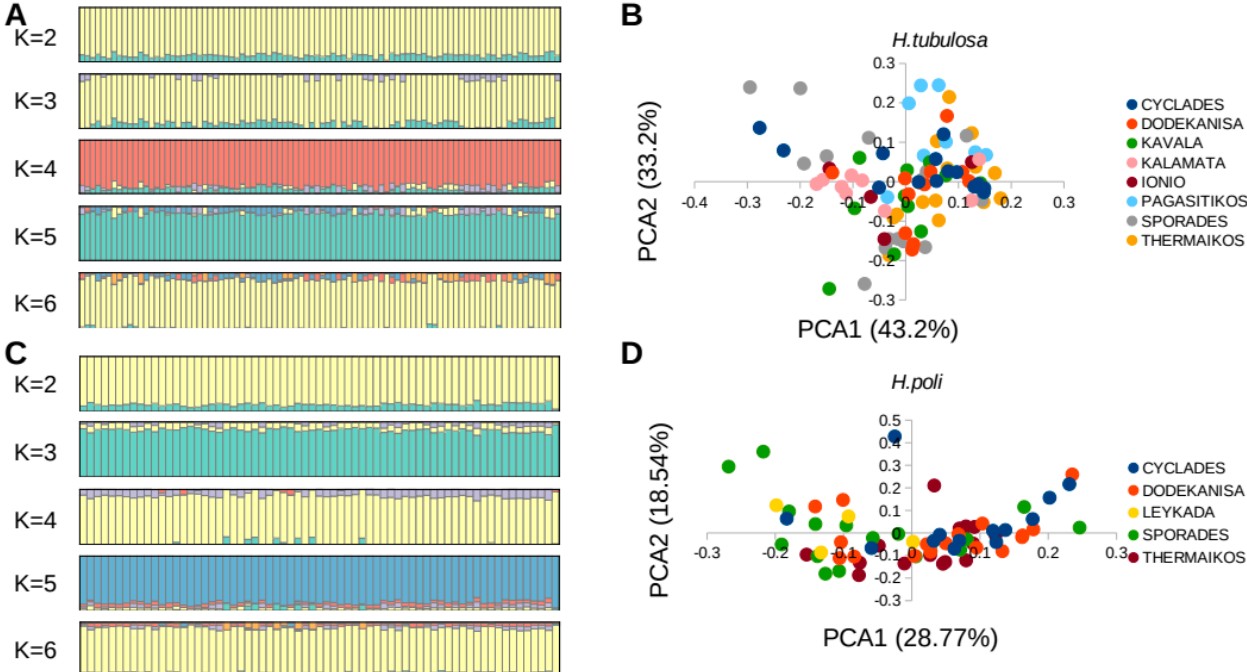

**Figure 3.** Evidence for a relevant panmixia of the *H. tubulosa* and *H. poli* in the Hellenic Seas (eastern Mediterranean basin). (**A**,**B**) Structure and PCA results for *H. tubulosa*. (**C**,**D**) Structure and PCA results for *H. poli*.

Using a model-based approach, the program ADMIXTURE assigned all individuals of both species, *H. tubulosa* and *H. poli*, to one cluster (Supplementary Information, Figure S2), suggesting high levels of admixture between the two lineages. NEWHYBRIDS v.1.1 software was run, assigning individual-level ancestry proportions to a single cluster (Supplementary Information, Figure S3).

*3.3. Coalescent Demographic Analyses*

Using six individuals from each species, the demographic history of the species in deep time was reconstructed based on coalescent analyses (with the pairwise sequential Markovian coalescent, PSMC). All twelve individuals showed essentially the same pattern (Figure 4). The effective population size (Ne) was approximately 60 thousand before the LGM, and dropped down to about 20 thousand during the last glacial period. As for the contemporary Ne, the LD method estimated a number of $4.43 \times 10^6$ and $4.74 \times 10^6$ for *H. tubulosa* and *H. poli*, respectively (Table 2). This pattern shows a rapid recovery of the populations in the area.

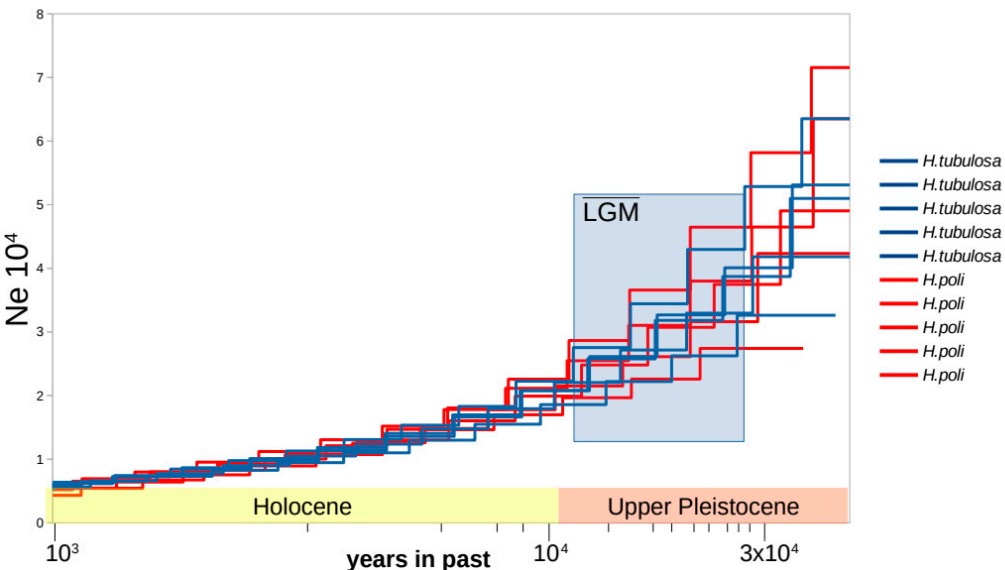

**Figure 4.** Demographic history of *H. tubulosa* (blue) and *H. poli* (red). The changes in ancestral population size of *H. tubulosa* and *H. poli* were inferred using the PSMC method (pairwise sequential Markovian coalescent). Time in history was estimated by assuming a generation time of 3 years and a mutation rate of $1.0 \times 10^{-8}$. LGM: Last Glacial Maximum.

**Table 2.** Prior distribution settings and parameter estimates for the fastsimcoal2 analyses, using $1 \times 10^6$ datasets simulated under scenario 4 (95% confidence intervals are shown for each of the parameters). Also, GONE analysis of contemporary population effective sizes for both *H. tubulosa* and *H. poli* lineages are reported. $N_{H.tubulosa}$: effective population size of *H. tubulosa*; $N_{H.poli}$: effective population size of *H. poli*; $N_{ANC}$: effective population size of historical population; $T_{DIV}$: time of divergence in generations (in parentheses in years); mig1: migration from *H. poli* to *H. tubulosa*; mig2: migration from *H. poli* to *H. tubulosa*.

| | Priors | Median | 95% CI (Low–High) | GONE |
|---|---|---|---|---|
| $N_{H.tubulosa}$ | $100$–$10^7$ | $5.6 \times 10^6$ | $3.59 \times 10^4$ –$9.32 \times 10^6$ | $4.43 \times 10^6$ |
| $N_{H.poli}$ | $100$–$10^7$ | $6.0 \times 10^5$ | $5.62 \times 10^4$ –$6.26 \times 10^5$ | $4.74 \times 10^6$ |
| $N_{ANC}$ | $100$–$10^7$ | $5.3 \times 10^4$ | $7.26 \times 10^3$–$9.32 \times 10^4$ | |
| $T_{DIV}$ | $100$–$10^7$ | $4.5 \times 10^3$ $(13.5 \times 10^3)$ | $3.32 \times 10^3$–$9.92 \times 10^3$ | |
| mig1 | $10^{-6}$–$10^{-3}$ | $0.002$ | $9.7 \times 10^6$–$4 \times 10^{-3}$ | |
| mig2 | $10^{-6}$–$10^{-3}$ | $8.5 \times 10^{-6}$ | $1.5 \times 10^6$–$4 \times 10^{-4}$ | |

Coalescent simulator results showed that the most likely demographic scenario was model 4, which supports the divergence of the two lineages around 4.5 thousand generations ago with a recent gene flow between themes (Table S4). This implies that the shrink of the populations occurred 13.5 thousands years ago, right after the Late Glacial Maximum (LGM), resulting in a decline of the population effective size to 5.3 thousand individuals (Table 2). Following that, a recent bidirectional gene flow with stronger direction from *H. poli* individuals to *H. tubulosa* ones was estimated to have a 0.2% migration rate.

## 4. Discussion

According to the presented results, the two studied holothurian species showed low levels of genetic differentiation in the Hellenic Sea. *H. tubulosa* and *H. poli* share the same niche in several Mediterranean locations with coexisting populations [24,26,54,55]. RADseq data provided evidence for the presence of admixed individuals and an overall panmixia of the two species. By including allopatric populations and performing demographic coalescent analyses, a recent gene flow was found between the two species, indicating a

possible collapse of the mating barriers to interbreeding. The interspecific gene flow was also evident when a model-based analysis was performed since all the individuals from both species were assigned to a single category, suggesting no clear reproductive isolation.

Studies in the literature show that hybridization is more frequently observed between species with similar morphological traits [54] and, therefore, in species on sympatry [16]. However, interbreeding between historically allopatric taxa reveals secondary contact as a consequence of a potential overlap of breeding periods. To this extent, species with a population demography that shrank and expanded during the Pleistocene glacial and interglacial periods may promote introgression.

In the case of holothurian, the data show that Pleistocene alterations of the species populations have experienced drastic changes in their population effective sizes during the last 60 thousand years. A given species introgression may have been encouraged by the competition for resources and habitat dependence during the glacial eras. This study reports that the rapid recovery of effective population sizes toward the Holocene period, along with the detection of recent gene flow, have led to hybridization among the two studied species. *H. tubulosa* and *H. poli* are two of the most abundant and commercially exploited holothurian species in the Mediterranean Sea [26]. As typical sediment-feeding organisms, they represent keystone species in benthic habitats [56,57]. The species are easily distinguished in situ thanks to the white spots of *H. poli*. However, when dried or preserved, their identification is rather difficult based on macroscopic morphological characters, as both species manifest increased phenotypic plasticity in relation to habitat features [58]. Accordingly, most authors argue that calcareous structures (i.e., ossicles) are the most prominent method for the precise taxonomy of holothurians [59–61]. However, a complete typology of ossicles is missing for most *Holothuria* species, and their skeletal elements are highly variable with respect to body region [62]. Moreover, hybridization between coexisting populations of sympatric species, such as those studied, adds confusion due to the presence of individuals with mixed morphological features [31,61]. Thus, the low levels of differentiation between the two *Holothuria* species in the present study may be due to an exchange of genetic variants through gene flow, resulting in a better response to adaptation [63]. To this extent, introgressive hybridization might take advantage of adaptive variants, allowing different lineages to exchange variants that have already undergone natural selection [63,64].

However, this exchange depends on the level of gene flow. The present data suggest an asymmetric pattern of gene flow with a direction from *H. poli* to *H. tubulosa*. This direction may imply that *H. tubulosa* carries more adaptive variants, which may be associated with introgression in the advancing species compared to the receding ones (e.g., [65]). This scenario, supported by the coalescence-based analysis, implies that *H. poli* has taken over from *H. tubulosa* since the LGM and reflects a directionally moving hybrid zone. Despite this, a given admixture zone may occur across many different contact areas, and so the reproductive isolation of the two lineages is still incomplete. All methods regarding genetic differentiation and population structure failed to separate clusters, as a high proportion of individuals of both species belonged to a single genetic group. However, *H. tubulosa* individuals from the Thermaikos Gulf showed significant divergence from most other geographic areas, implying a complex barrier [66]. This may also be attributed to a methodological bias, such as the relatively low number of individuals from the no differentiation areas and the relative difference in the number of RAD loci and variants. This might be due to technical issues (DNA quality and/or applied enzymes) or to the choice of the cross-reference genomes for SNP calling. Similarities in demographic history may explain the observed similar genomic diversity between the two species. For instance, the effective population size of both species probably declined during the Pleistocene glacial periods, and then, recovered to similarly high contemporary effective population sizes. Though the relatively small number of analyzed holothurians in some geographical areas may have led to ambiguity or misinterpretation of the outcomes, recent studies using high-throughput sequencing argue that population genetics can be performed with small sample

sizes [67,68]. However, other factors, including dispersal ability, demographic history, and overall population dynamics of the studied species, should be considered in any effort to determine an ideal sampling scheme [69].

## 5. Conclusions

Concluding the presented results suggests a genetic admixture upon secondary contact with a recent gene flow that supports cryptic speciation and high habitat connectivity between the two holothurian lineages. The observed absence of genetic differentiation may be subject to the exchange of adaptive traits through promoted introgression. The extremely recent estimated time of divergence, around 13.5 thousand years ago, supports the striking result of admixture between the two lineages with a more recent gene flow, suggesting a complex and incomplete reproductive isolation between *H. tubulosa* and *H. poli*.

**Supplementary Materials:** The following supporting information can be downloaded at https://www.mdpi.com/article/10.3390/su151511493/s1, Table S1: Filtering steps that were carried out to generate the final high-quality SNP dataset starting from the raw SNPs output by the GATK pipeline. The number of retained SNPs after each step is reported. GQ: Genotype quality; DP: Genotype depth of coverage; IGR: Individual genotyping rate; maxDP: Depth of coverage (twice the mean depth of coverage of the raw dataset); MAF: Minimum Allele Frequency; Table S2: Genetic indices of the *H.tubulosa* and *H.poli* geographical populations as calculated in Sambar (ref). Numbers in parantheses indicate number of individuals; Table S3: Structure results of the likelihood value of the different K values and and ΔK as implementing in Evanno method; Table S4: The Akaike information criterium (AIC) of each of the five tested demographic scenarios. Maximum Likelihood values are also reported for each model. Also, the weighted AIC values are reported for the close 3 scenarios regarding their original AIC values; Figure S1: PCA analysis based on both species individuals. Blue dots: *H.tubulosa*; Red dots: *H.poli*; Figure S2: Admixture histogram of the detected clusters and assignment of the individual-level ancestry proportions from each cluster; Figure S3: NEWHYBRIDS histogram of the posterior probabilities of the *H.tubulosa* and *H.poli* individuals. Among the 157 individuals analysis assigned them to single category (blue color). Reference [70] is cited in the Supplementary Materials.

**Author Contributions:** Conceptualization, D.V.; methodology, C.A. (Chryssanthi Antoniadou), D.V., A.E., G.A.G. and J.S.; software, G.A.G.; validation, C.A. (Chryssanthi Antoniadou), A.E., G.F. and G.A.G.; formal analysis, G.A.G. and J.S.; resources, D.V.; data curation, C.A. (Chrysoula Apostologamvrou), D.V. and J.S.; writing—original draft preparation, G.A.G.; writing—review and editing, C.A. (Chryssanthi Antoniadou), D.V., G.F. and A.E.; supervision, D.V.; project administration, D.V.; funding acquisition, D.V. All authors have read and agreed to the published version of the manuscript.

**Funding:** This work was implemented in the framework of the project entitled "Exploitation and management of sea cucumber fisheries (*Holothuria* spp.): processing (food and biotech products) and safeguarding of stocks" with MIS 5010720 which was funded from the European Union, European Maritime and Fisheries Fund, in the context of the Operational Programme "Maritime and Fisheries 2014–2020".

**Institutional Review Board Statement:** Not applicable.

**Informed Consent Statement:** Not applicable.

**Data Availability Statement:** The datasets generated during and/or analyzed during the current study are available in the NCBI repository (accession number: PRJNA981972).

**Conflicts of Interest:** The authors declare no conflict of interest. The funders had no role in the design of the study; in the collection, analyses, or interpretation of data; in the writing of the manuscript; or in the decision to publish the results.

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
