# Peer review of "Admixture of Holothurian Species in the Hellenic Seas (Eastern Mediterranean) as Revealed by RADseq"

_sustainability, doi:10.3390/su151511493_

Round 1
Reviewer 1 Report
I have reviewed the paper by Gkafas et al., which analyzed the population genomics of two closely related holothurian species based on SNP data.
Overall, the manuscript needs a thorough check of English grammar and style, especially in the first two paragraphs of the Introduction, but also in the Discussion. Taking all the analyses in consideration, it seems to me that these two lineages, actually constitute a single species: the Structure results for K=2 show very similar assignment probabilities to all individuals for the same cluster, which to me suggest that the optimal K=1. In addition, the admixture results also suggest that both species belong a single cluster. I suggest that the authors apply a species delimitation approach to test if these two lineages actually constitute different species.
My other major criticism is the set of models compared with coalescent simulation. Based on the demographic analyses, there has been population decline in the ancestral and extant populations. However, the simulated models, it looks like a constant population size was assumed, which could have impacted divergence time estimates. I think it is possible to implement changes in population size in fastsimcoal2. In addition, even if model 4 has the lowest AIC, it is still very similar to the AIC values of models 2 and 3. I think there is similar support for all 3 models. I suggest calculating and interpreting Akaike weights for each model.
The authors should address these two major issues in a revised version taking also into account the following minor points and typos:
Ln 12. Replace “species” with “population”
Ln 22. Replace “shrink” with “decline”
Ln 29. Delete “to natural”.
Ln 30. Replace “with respect to” with “driven by”
Ln 42. Replace “historial allele frequencies” with “ancestral alleles”
Ln 52. Replace “somewhat” with “limited”
Ln 71. Describe these morphological features.
Ln 75. Replace “following” with “followed”
2. 1 Sampling. Were specimens deposited in a collection? Any collecting permits?
Ln 121. “depth”
Ln 131. Replace “repeats” with “replicates”
Ln 132. Why assuming a max K=6? Given the number of sampled areas, K could be as large as 9.
Ln 209. Insert “differentiation” between “significant” and “was”
Ln 214. It could be interesting to evaluate an isolation-by-distance pattern.
Ln 218. Replace “somewhat” with “apparent”
Ln 224. Delete “holothurian”
Ln 233. Insert “(Table S4)”
3.3 Coalescent analyses (“population decline”). The demographic results are described in the following paragraph. The first paragraph should be the second paragraph, and viceversa.
Ln 235. Actually, migration is bidirectional, but stronger in one direction.
Discussion. The first paragraph suggests that both lineages are a single species.
Ln 305. How do you know these taxa were allopatric in the past?
Ln 307. Which are the breeding periods?
Ln 312. How do you know the population decline started 60 kya?
Ln 312. I do not understand this sentence.
Ln 330. This paragraph about the possible adaptive value of the migration is very speculative.
2.14.0.0
Overall, the manuscript needs a thorough check of English grammar and style, especially in the first two paragraphs of the Introduction, but also in the Discussion.
Author Response
Reviewer_1
I have reviewed the paper by Gkafas et al., which analyzed the population genomics of two closely related holothurian species based on SNP data.
Comment_1
Overall, the manuscript needs a thorough check of English grammar and style, especially in the first two paragraphs of the Introduction, but also in the Discussion.
Response:
The ms has been revised thoroughly regarding the syntax and grammar.
Comment_2
Taking all the analyses in consideration, it seems to me that these two lineages, actually constitute a single species: the Structure results for K=2 show very similar assignment probabilities to all individuals for the same cluster, which to me suggest that the optimal K=1.
Response:
We thank the reviewer for this comment. Actually, in our data set there isn't much/or any population structure expected for these guys tested, since according to our results (PSMC and GONE) they expanded over in a relatively narrow geographic range (Greek Seas) fairly recently after the bottleneck during the LGM.
One of the profound signatures of weak or non-existent population structure is the admixture at clusters K=2 and higher, where there is no definitive assignment of individuals to particular clusters. Unfortunately, the Evanno method cannot work with K=1, so that’s why we present results for K=2 and higher; goodness of fit data.
Comment_3
In addition, the admixture results also suggest that both species belong to a single cluster. I suggest that the authors apply a species delimitation approach to test if these two lineages actually constitute different species.
Response:
Species verification is undoubted & was assessed through extensive-attentive morphological identification by specialists in the field (Dr. Antoniadou and Prof. Vafidis: co-authors of the manuscript), and usage of mtDNA COI sequences were also checked by molecular experts (by Prof. Exadactylos and Dr. Gkafas: co-authors of the paper). In fact, this is the core finding of the paper; different species with similar admixture genetic indices within the holothurian genus.
Comment_4
My other major criticism is the set of models compared with coalescent simulation. Based on the demographic analyses, there has been population decline in the ancestral and extant populations. However, the simulated models, it looks like a constant population size was assumed, which could have impacted divergence time estimates. I think it is possible to implement changes in population size in fastsimcoal2.
Response:
Reviewer’s comment regarding a potential bias of time of divergence estimation when assuming constant population sizes using ABC methods, might be true in theory. However, in studies of non-model organisms, where knowledge of past demographic events is limited or non-existent, it is common and more “safer” not to use complex models (see Roux et al, 2013, and also Laurent Excoffier suggestions in fastsimcoal2 manual and his relative publication). On the other hand, in our data set via PSMC analysis we have clear evidence of a decline of both species populations during and especially after the LGM (with a rapid decrease in Ne), which is in full concordance with fastsimcoal2 estimation of the divergence time. Thus, the estimated time is NOT biased by using constant sizes in our case. On top of that, using fastsimcoal2, our main purpose and tendency was to show and test for any potential migration pattern that exists especially when secondary contacts are evident.
Comment_5
In addition, even if model 4 has the lowest AIC, it is still very similar to the AIC values of models 2 and 3. I think there is similar support for all 3 models. I suggest calculating and interpreting Akaike weights for each model.
Response:
Relative calculations were added to Table S4. AIC weighted values definitely support the 4th model.
Comment_6
The authors should address these two major issues in a revised version taking also into account the following minor points and typos:
Response:
We hope that we delivered the answers to the reviewer’s concerns with adequate argumentative explanatory manner.
Minor comments
minor_comment_1
Ln 12. Replace “species” with “population”
Response:
done
minor_comment_2
Ln 22. Replace “shrink” with “decline”
Response:
done
minor_comment_3
Ln 29. Delete “to natural”.
Response:
done
minor_comment_4
Ln 30. Replace “with respect to” with “driven by”
Response:
done
minor_comment_5
Ln 42. Replace “historical allele frequencies” with “ancestral alleles”
Response:
done
minor_comment_6
Ln 52. Replace “somewhat” with “limited”
Response:
done
minor_comment_7
Ln 71. Describe these morphological features.
Response:
The morphological features are thoroughly described in the ‘Discussion’ section.
minor_comment_8
Ln 75. Replace “following” with “followed”
Response:
done
minor_comment_9
2. 1 Sampling. Were specimens deposited in a collection? Any collecting permits?
Response:
Samples are stored in Hydrobiology-Ichthyology Laboratory, of the Department of Ichthyology and Aquatic Environment, (University of Thessaly) following ISO:9001 instructions & GLP protocol.
minor_comment_10
Ln 121. “depth”
Response:
done
minor_comment_11
Ln 131. Replace “repeats” with “replicates”
Response:
done
minor_comment_12
Ln 132. Why assuming a max K=6? Given the number of sampled areas, K could be as large as 9.
Response:
Reviewer is right regarding the max K. However, we ran an extra analysis based on K=9, and had no alterations in our results.
minor_comment_13
Ln 209. Insert “differentiation” between “significant” and “was”
Response:
done
minor_comment_14
Ln 214. It could be interesting to evaluate an isolation-by-distance pattern.
Response:
This was not the purpose of the paper, since we don’t have such evidence according to our findings & null hypotheses.
minor_comment_15
Ln 218. Replace “somewhat” with “apparent”
Response:
done
minor_comment_16
Ln 224. Delete “holothurian”
Response:
done
minor_comment_17
Ln 233. Insert “(Table S4)”
Response:
done
minor_comment_18
3.3 Coalescent analyses (“population decline”). The demographic results are described in the following paragraph. The first paragraph should be the second paragraph, and vice versa.
Response:
done
minor_comment_19
Ln 235. Actually, migration is bidirectional, but stronger in one direction.
Response:
We thank reviewer for the comment. We corrected the text accordingly.
minor_comment_20
Discussion. The first paragraph suggests that both lineages are a single species.
Response:
We don’t agree that this part reports that both lineages are a single species. We discuss the notion, as it has been recorded in a numerous set of studies reporting hybridization; our findings provide clues on the reproductive behavior and dispersal capabilities of the two species. Yet again we feel that this is our core finding to be discussed & cited in the future concerning the status of the two species and their genetic equilibrium.
minor_comment_21
Ln 305. How do you know these taxa were allopatric in the past?
Response:
We thank reviewer for the comment. We deleted the “such as the studied holothurians” as it was written by mistake.
minor_comment_22
Ln 307. Which are the breeding periods?
Response:
Breeding periods are mixed between them and it has been reported that both species share an extensive period that lasts several months of the same year (personal communication Dr. Vafidis - Dr. Antoniadou). In fact, maybe this is an adaptive response to pressure… remains to be checked in the near future.
minor_comment_23
Ln 312. How do you know the population decline started 60 kya?
Response:
According to the results of the PSMC analysis. We report this in a detailed manner in the ‘Results’ section.
minor_comment_24
Ln 312. I do not understand this sentence.
Response:
We tend to argue that a given introgression between species may be encouraged by the competition for resources and habitat dependence during the glacial eras. We re-phrased the sentence as such.
minor_comment_25
Ln 330. This paragraph about the possible adaptive value of the migration is very speculative.
Response:
We use relative references that support the adaptive introgressive variants, where given variants are introduced from the “donor” species into the “recipient” species by hybridization, in such a way that offspring are more adaptive to pressure and present more fitted reproductive strategy; evolutionary functioning theory in other words.
Reviewer 2 Report
I found this manuscript very interesting and innovative, based on a well-conducted study with sufficient samples to give a first contribution to this topic.
Please double-check the italics of scientific names within the manuscript.
Scenario's Figure: please add a caption. Moreover, a colour image could be improve the clarity of this scheme.
Lines 344-346: in the present form it's not clear if the authors refers on overfishing for their samples or the references ones. Introduction section don't report information about this problem in the Mediterranean Sea's studied area. Please add support to this period also in introduction section exposing the theme, or avoid speculations in discussion section about it.
The confirmation of the allopatric origin of the Holothurian population (even if the sample number is not huge), basing on a genetic investigation.
The novelty of this study is based on the applied methods which are describing several taxa relationships in the last years.
The admixture between the two species H. tubulosa and H. poli, based on their genetic diversity was investigated for the first time in the studied area.
The experimental design was in my opinion well-pondered, and also the proposed scenarios are reasonable.
Best regard
The reviewer
Please revise the English language by yourself before proofreading.
Author Response
Reviewer_2
I found this manuscript very interesting and innovative, based on a well-conducted study with sufficient samples to give a first contribution to this topic.
Comment_1
Please double-check the italics of scientific names within the manuscript.
Response:
Italics and scientific names are double-checked.
Comment_2
Scenario's Figure: please add a caption. Moreover, a colour image could be improve the clarity of this scheme.
Response:
We don’t understand reviewer’s comment, since there is a caption present on Fig. 2 in the text.
Comment_3
Lines 344-346: in the present form it's not clear if the authors refers on overfishing for their samples or the references ones. Introduction section don't report information about this problem in the Mediterranean Sea's studied area. Please add support to this period also in introduction section exposing the theme, or avoid speculations in discussion section about it.
Response:
We thank reviewer’s comment. We deleted the overfishing as a potential factor, since we don’t have enough published data for that, although is totally true through personal communications with fishermen.
Comments_4
The confirmation of the allopatric origin of the Holothurian population (even if the sample number is not huge), based on genetic investigation.
The novelty of this study is based on the applied methods which are describing several taxa relationships in the last years.
The admixture between the two species H. tubulosa and H. poli, based on their genetic diversity was investigated for the first time in the studied area.
The experimental design was in my opinion well-pondered, and also the proposed scenarios are reasonable.
Response:
We thank reviewer’s comment concerning our null hypothesis & our presented result design.
Reviewer 3 Report
This study utilizes RAD sequencing technology to analyze the genetic diversity of two closely related Holothurian species living in sympatry with different subgenera. The findings provide the evidence of admixture in mixed populations of Holothuria (Holothuria) tubulosa and Holothuria (Roweothuria) poli in various areas of the Hellenic Seas. The results support the notion that allopatric origin plays a role in ecological speciation and admixture upon secondary contact. Overall, the manuscript needs to be improved before it could be considered for publication.
My comments are listed below.
Materials and methods
In the section “Sampling”, I did not find any information about external features of the sample, such as the length and weight of 180 sea cucumbers, and water environmental conditions (temperature, pH, salinity) and so on. I suggest to add these information in this section.
Result: Where is Figure 2? Maybe the five Scenario is Figure 2, please add figure caption.
Author Response
Reviewer_3
This study utilizes RAD sequencing technology to analyze the genetic diversity of two closely related Holothurian species living in sympatry with different subgenera. The findings provide the evidence of admixture in mixed populations of Holothuria (Holothuria) tubulosa and Holothuria (Roweothuria) poli in various areas of the Hellenic Seas. The results support the notion that allopatric origin plays a role in ecological speciation and admixture upon secondary contact.
Comment_1
Overall, the manuscript needs to be improved before it could be considered for publication.
Response:
We thank the reviewer for the comment. We have improved the document following all reviewers’ comments/suggestions.
My comments are listed below.
Materials and methods
Comment_2
In the section “Sampling”, I did not find any information about external features of the sample, such as the length and weight of 180 sea cucumbers, and water environmental conditions (temperature, pH, salinity) and so on. I suggest to add this information in this section.
Response:
We thank the reviewer for the comment, but since we don’t execute any fitness correlation analysis we didn’t include any length/weight info of the studied specimens. However, sampling strategy was that of a true randomized approach using a quadrat plot. Moreover, we didn’t have the chance to record any of the environmental conditions.
We solely focused our work to the genomic null hypothesis scheme we implemented, and as always we deposited our raw RAD-seq data to reference banks (accession numbers are referred to the text) for others to use in the future.
Comment_3
Result: Where is Figure 2? Maybe the five Scenario is Figure 2, please add figure caption.
Response:
We don’t understand reviewer’s comment. Figure 2 refers to the 5 tested scenarios, which is included in the ms, and it has a relevant caption.
Round 2
Reviewer 1 Report
The Akaike weights are wrong. See below the correct values. Based on these weights, which are low and very similar among models 2, 3, and 4, you should be doing model averaging using these 3 models to estimate model parameters.
2.14.0.0
2.14.0.0
2.14.0.0
2.14.0.0Author Response
We thank the reviewer for the correction. Indeed when we used the formula for the weighted AIC (exp(-0.5 * ΔAIC), due to the mistype of the formula in excel we didn't use the delta AIC, but the original AIC values. We re-calculate the delta AIC and weighted AIC and we are in concordance with the reviewer.
According to Burnham and Anderson (2002), as a rule of thumb, a ΔAIC < 2 suggests substantial evidence for the model. In this case, indeed model 2, 3 and 4 meet the criteria.
Further we ascribe a probability to the models via
p = exp (-ΔAIC/2),
, which provides a relative probability (Kenneth P. Burnham and David R. Anderson, 2004 - DOI: 10.1177/0049124104268644).
Finally, Model averaging is then calculated by
mod_avg = p x weightedAIC
as shown in Table S4.
The model 4 is slightly better supported than model 2. However, it is still not clear which of these two is supported. For this reason we performed a F-test comparison of the two models (2 and 4) based on the sum of squares of the fastsimcoal output. Results show a better support for model 4 (p<0.001, F=7.84). Thus, we tend to follow our initial argument that model 4 is the most favorable. Relevant info added to the Supplementary (M&Ms section).
